# *LanEvil*: Benchmarking the Robustness of Lane Detection to Environmental Illusions

## ABSTRACT

Lane detection (LD) is an essential component of autonomous driving systems, providing fundamental functionalities like adaptive cruise control and automated lane centering. Existing LD benchmarks primarily focus on evaluating common cases, neglecting the robustness of LD models against environmental illusions such as shadows and tire marks on the road. This research gap poses significant safety challenges since these illusions exist naturally in real-world traffic situations. For the first time, this paper studies the potential threats caused by these environmental illusions to LD and establishes the first comprehensive benchmark *LanEvil* for evaluating the robustness of LD against this natural corruption. We systematically design 14 prevalent yet critical types of environmental illusions (*e.g.*, shadow, `reflection`) that cover a wide spectrum of real-world influencing factors in LD tasks. Based on real-world environments, we create 94 realistic and customizable 3D cases using the widely used CARLA simulator, resulting in a dataset comprising 90,292 sampled images. Through extensive experiments, we benchmark the robustness of popular LD methods using *LanEvil* , revealing substantial performance degradation (-5.37% Accuracy and -10.70% F1-Score on average), with shadow effects posing the greatest risk (-7.39% Accuracy). Additionally, we assess the performance of commercial auto-driving systems OpenPilot and Apollo through collaborative simulations, demonstrating that proposed environmental illusions can lead to incorrect decisions and potential traffic accidents. To defend against environmental illusions, we propose the Attention Area Mixing (AAM) approach using hard examples, which witness significant robustness improvement (+3.76%) under illumination effects. We hope our paper can contribute to advancing more robust auto-driving systems in the future. Part of our dataset and demos can be found at the anonymous website.

## CCS CONCEPTS

• **Security and privacy**; • **Computing methodologies → Machine learning**;

## KEYWORDS

Lane Detection, Environmental Illusion, Robustness Benchmark

*ACM MM, 2024, Melbourne, Australia*

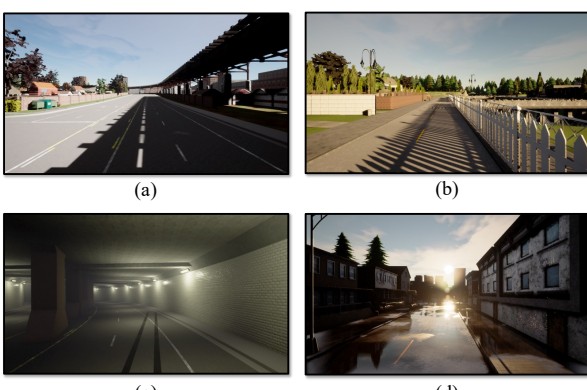

**Figure 1: Illustration of naturally existing yet overlooked environmental illusions (*e.g.*, shadow). Perception of these patterns that objectively exist in such a way could cause misinterpretation of their actual nature leading to wrong lane recognition.**

## 1 INTRODUCTION

Lane detection aims to identify the location of lane lines or road edges [24, 41–43, 63], which now serves as the foundation for many driving assistant functions in the real-world auto-driving vehicles [21], such as lane centering and adaptive cruise control.

Though demonstrating promising results on datasets of common traffic cases (*e.g.*, TuSimple [55], CULane [41]), the robustness of LD models on cases containing environmental illusions remains unexplored. In real-world traffic cases, there exists a range of *naturally existing yet overlooked environmental illusions*, such as shadow and reflection in Figure 1. These environmental illusions are natural to human perception. However, perception of these deceiving patterns that objectively exist in such a way could bring natural corruption and cause misinterpretation of their actural nature leading to wrong lane recognition. This sparsity of research presents a severe risk to the safety of auto-driving systems, as it increases their vulnerabilities and poses risks to human lives and properties [58]. Considering the safety-critical nature of autonomous driving, it is of paramount importance to rigorously evaluate LD robustness on such environmental illusions before deployment.

In this paper, we take the first step in studying LD model robustness towards the environmental illusions. In contrast to the common corruption datasets [12, 20] (*e.g.*, snow) that directly perturb the overall clean input images with noises, environmental illusions target adding case-specific and naturally-looking modifications to some regions of the case. Thus, environmental illusions should be considered as a new type of natural corruption robustness problem. Though naturally existing in practice, it is still difficult to directly collect such large-scale real-world images containing different types of environmental illusions. Simulation testing, renowned for its cost-efficiency and reproducibility, has become a widely adopted

alternative to real-world experimentation, especially within the realm of autonomous driving research [1, 16, 47]. Therefore, we adopt the commonly used simulation testing pipeline [28, 29, 56] to collect synthetic images [45, 46, 49]. In particular, we utilize the commonly used CARLA simulator [13] to intricately design 3D traffic environments; we then physically perturb the visual properties of the target objects (*e.g.*, roads) with our proposed environmental illusions and finally render the corrupted images.

Based on the pipeline, we rigorously analyze the critical influence factors in LD scenarios including *Dynamic Objects*, *Static Facilities*, and *Environmental Conditions*, and systematically design 14 illusion types with 5 severity levels that collectively address a wide spectrum of real-world environmental illusions on the road and propose the *LanEvil* LD robustness evaluation benchmark. Overall, our benchmark encompasses 94 cases with editable 3D environments and a 90,292 sampled image dataset with 40,000 clean training images and 50,292 test images. Leveraging *LanEvil* , we conducted extensive experiments to benchmark the robustness of commonly adopted LD models, where we observed severe performance degeneration (-5.37% Accuracy and -10.70% F1-Score on average). Notably, the shadow effect caused the most considerable reduction (-7.39% Accuracy). To improve model robustness against environmental illusions, we propose the Attention Area Mixing (AAM) approach, a novel noise defense baseline leveraging hard examples, achieving a 3.76% improvement over vanilla detectors towards these illusions. To better demonstrate the potential of our *LanEvil*, we conducted joint simulation experiments on commercial autonomous driving systems (OpenPilot [9] and Apollo [3]), where we observed incorrect decisions resulting in car accidents. Finally, we also conducted case studies on real-world scenarios containing environmental illusions, which demonstrate the threats in practice. We hope this paper will raise awareness regarding potential security threats in autonomous scenarios. Our **contributions** are as follows:

- As far as we know, we are the first to study the influence of environmental illusions on the robustness of LD models (an essential component in auto-driving).
- We build the *LanEvil* benchmark, which contains 14 typical environmental illusions, 90,292 images, and 94 editable 3D cases supporting user customization.
- We introduce the Attention Area Mixing (AAM) approach, leveraging hard examples to surpass existing noise defense techniques in addressing environmental illusions.
- We conduct extensive experiments on commonly used LD models, and commercial autonomous driving systems, which substantiates its real-world threats.

## 2 RELATED WORK

### 2.1 Lane Detection

Lane detection addresses the identification of lane lines or road edges. Currently, deep learning-based LD methods have emerged as the predominant paradigm, harnessing their ability to learn complex features and patterns. In general, the mainstream LD methods can be divided into the following five categories as *segmentation-based methods* [37, 39, 41] that treat LD as a pixel-level classification task; *keypoint-based methods* [25, 44, 57] that identify critical points for LD and subsequently group these key points into lane line

instances; *anchor-based methods* [30, 52, 53] that utilize predefined anchor points to efficiently identify and locate lane boundaries in images; *row-wise classification methods* [23, 42, 43, 60] that estimate the cell that most probably contains a lane line for each row and repeat this process for each lane; and *parameter-based methods* [8, 15, 36, 54] that model the LD task as a curve fitting problem.

In this paper, we will then benchmark and evaluate the robustness of all the above types of LD methods.

### 2.2 Lane Detection Datasets

The advancement of LD is closely tied to high-quality datasets under various traffic cases. Early datasets are relatively simple, such as the CalTech dataset [2], which contains 1,224 frames in common urban streets without weather changes. In contrast, VPGNet [27] introduces more complexity, offering 20,000 images featuring intricate urban traffic cases; TuSimple [55] primarily concentrates on the annotated lane under highway scenes. Besides, some other datasets introduce more diverse traffic conditions. For example, CU-Lane [41] includes over 130,000 images, with approximately 72.3% of the dataset featuring challenging cases such as traffic crowds and dazzling light; BDD-100K [61] covers diverse lighting conditions and 6 extreme weather types; in LLAMAS [4], the count of marked lane pixels is sparse and varies with marking distance and position; in addition, CurveLane [59] places emphasis on curved lanes.

Besides the above common datasets, the **robustness benchmarks** that study natural corruption in LD scenarios are rare. Some pioneering benchmarks are devoted to assessing the robustness of image classification [20] and object detection [20] against common perturbations such as blur, weather conditions, and digital corruption. Recently, some studies have proposed to investigate the perception robustness of autonomous driving against common corruption [12, 26]. However, these works primarily focus on general 3D perception tasks such as detection and segmentation, and the generated corruption (*e.g.*, motion blur) is not specially designed for LD. Moreover, there also exist some studies that generate specially designed lane-like adversarial attacks for autonomous driving [6, 48], which is out of the scope of this paper.

To sum up, though existing LD datasets have achieved notable milestones in assessing the performance of LD methods, there still exist no systematic investigations on LD scenario-related corruption, such as shadows and reflections. Our *LanEvil* aims to bridge this gap by providing a comprehensive dataset for LD corruption robustness evaluation. *The detailed comparison of* LanEvil *and other datasets are shown in Supplementary Materials.*

## 3 THE *LANEVIL* DATASET

### 3.1 Problem Definition

**Lane detector.** A lane detector $f_{\theta}(\mathbf{I}) \rightarrow \mathbf{loc} \in \mathbb{N}^K$ with parameters $\theta$, which takes an image $\mathbf{I} \in [0, 255]^3$ as input, outputs $K$ lane line locations as **loc**. The formulation of the training is as follows:

$$\min_{\theta} \mathbb{E}_{(\mathbf{I}, \mathbf{loc}_{gt}) \sim \mathbb{D}} \mathcal{L}(f_{\theta}(\mathbf{I}), \mathbf{loc}_{gt}), \qquad (1)$$

where $\mathbb{D}$ denotes the dataset, and $\mathcal{L}(\cdot)$ is the loss function that measures the difference between the output of the lane detector $f$ and the ground truth $\mathbf{loc}_{gt}$.

**Environment.** In practice, the autonomous vehicle first perceives the real-world scenario environment $\Phi$ via the sensor and then projects/renders the 3D objects into the 2D image $\mathbf{I} = R(\Phi)$ as the input, where $R(\cdot)$ is the environmental sampling function. Specifically, the environment highly related to the LD scenario can be roughly divided into the static facilities $\mathbb{S} = \{\mathbf{s}_1, \mathbf{s}_2, ..., \mathbf{s}_n\}$ (*e.g.*, roads, fences) and dynamic objects $\mathbb{X} = \{\mathbf{x}_1, \mathbf{x}_2, ..., \mathbf{x}_m\}$ (*e.g.*, pedestrians, vehicles). In addition, the environmental conditions $\mathbf{C}$, such as weather and lighting, can also pose influences on the environment. Therefore, the environment $\Phi$ should be defined as

$$\Phi = (< \mathbb{S}, \mathbb{X} >, \mathbf{C}). \tag{2}$$

The input $\mathbf{I}$ of LD should be rendered from the environment with specific sets of static/dynamic objects with certain conditions as

$$\mathbf{I} = R(< \mathbb{S}, \mathbb{X} >, \mathbf{C}). \tag{3}$$

**Environmental illusions on LD models.** Natural changes/ modifications of the aforementioned static facilities, dynamic objects, and environmental conditions would bring certain corruption to the rendered input $\mathbf{I}$ and cause the performance degeneration of LD models. In particular, to generate environmental illusions, we directly modify the attributes of each parameter under the environment $\Phi$ in Equation 2 to get $\hat{\Phi}$. Then, the rendered image $\hat{\mathbf{I}} = R(\hat{\Phi})$ is slightly perturbed and contains environmental illusions in specific regions. Therefore, the performance $f_{\theta}(\hat{\mathbf{I}})$ of LD model may decrease. In this paper, we mainly design single-factor changes and generate the corresponding environmental illusions.

## 3.2 Environmental Illusion Design

As shown in Figure 2, our benchmark encompasses four environmental illusion categories with 14 types. We then illustrate the design of each category.

❶ **Road Damage.** The road damage corresponds to the influences brought by the perturbations added to the static facilities $\mathbb{S}$ on the road. Here, we design four typical types of environmental illusions including `Road Crack`, `Road Repair`, `Tire Marks`, `Guard Rail`. In particular, `Road Crack` illusion encompasses common forms of cracks, ranging from minor transverse and longitudinal cracks to mesh-like fissures. The repaired regions are visually obvious and some of them exhibit linear patterns that are similar to lane markings, and we term this `Road Repair` illusion. We also design the `Tire Marks` illusion which depicts the case of emergency braking or collisions.Moreover, we consider the potential misidentification of `Guard Rail` as lane markings, which can affect the detection of road edges.

Given an existing static object $\mathbf{s}_i$ (*e.g.*, road) in the environment $\Phi$, we modify the appearance attributes of object $\mathbf{s}_i$ by physically adding the above specific perturbation type on specific regions and obtain the perturbed object as $\hat{\mathbf{s}}_i = \mathbf{s}_i + g(\boldsymbol{\delta})$, where $g(\cdot)$ is the perturbed function and $\boldsymbol{\delta}$ denotes the severity levels and perturbed mask. Additionally, we add new object $\mathbf{s}_j$ on the roadside in $\Phi$, such as guard rails. Therefore, the environmental illusions caused by Road Damage can be formulated as perturbing or adding the static facilities in the environment as $\hat{\mathbb{S}} = \{\hat{\mathbf{s}}_1, ..., \hat{\mathbf{s}}_i, \mathbf{s}_{i+1}, \cdots, \mathbf{s}_{i+N}\}$, where $N$ is the number of added objects.

❷ **Traffic Obstruction.** This category of illusion corresponds to the influence of dynamic objects $\mathbb{X}$ on the roads. Traffic participants constitute a pivotal element in the realm of autonomous driving, often necessitating stringent safety measures. In this paper, we mainly focus on three classical types of traffic participants `Pedestrian`, `Vehicle`, and `Bicycle`. These participants could obstruct the view of lane lines and influence the performance of LD. In addition, we also generate different quantities of participants to simulate traffic flow with different levels of complexity.

In detail, for each specific case, we generate $n$ instances $\mathbf{x}_i$ of the specific illusion type. Different from Road Damage, we set the default $\mathbb{X}$ in the environment as $\varnothing$ and add participants at different regions on the roads. Therefore, the environmental illusions caused by Traffic Obstruction can be formulated as adding extra $m$ participants in the environment as $\hat{\mathbb{X}} = \{\mathbf{x}_1, \mathbf{x}_2, \mathbf{x}_3, \cdots, \mathbf{x}_m\}$.

❸ **Shadow.** The environmental conditions $\mathbf{C}$, such as weather and lighting, can affect the visual appearance of the roads and related objects resulting in extra illusions. One influential factor is the lighting condition changes over time, which would project different shadows on the road. These shadows (*e.g.*, shadows of the fence or streetlight) have patterns that are similar to lane lines, which could cause a decrease in LD performance. For example, at nightfall, the light angle is comparatively huge resulting in a larger shadow area; the light intensity at noon is high while the angle is small causing clearer shadow edges. Considering the above circumstances, we design four types of environmental illusions for shadows including `Streetlight`, `Fence`, `Rail`, and `Wire`. Specifically, `Streetlight` and `Fence` cast elongated shadows on the road when the sun is at a low elevation angle. Meanwhile, we also devise a type of shadow deception caused by the translucent parts of `Rail`. By adjusting the lighting angles, rails can project bright lines on the road that closely resemble lane markings in both color and shape, creating a highly deceptive visual effect. Finally, we observe a similar effect of `Wire`. Both the opaque parts of the shadow and the translucent areas between two power lines could be mistaken for lane markings.

Based on the above analysis, we generate the shadow illusion by altering the lighting factor in the simulation environment. The perturbation images $\mathbf{I}^{(l,a)}$ can be calculated by Equation 3 with hyper-parameter $(l, a)$ luminance and angle.

❹ **Reflection.** Another environmental condition weather changes, such as rains, would create water puddles on the roads, which can reflect the light and influence the accurate capture of lane lines. Images captured by autonomous vehicles at backlit angles are significantly distorted due to the impact of reflection. Based on the above observation, we design three types of illusion including `Sunlight Reflection`, `Streetlight Reflection`, and `Vehicle Reflection`. In particular, when the vehicle travels into the sun, intense `Sunlight Reflection` can blur lane markings, especially white lines and dashed lines. Moreover, after heavy rain, this effect becomes more pronounced, as the reflection from water on the road makes lane markings invisible in the images. Similarly, nighttime `Streetlight Reflection` also strongly affects the recognition of road lane markings. To make the illusion more severe and practical, we also consider the two most common colors of streetlights, *i.e.*, white and yellow, coincidentally matching the common colors of lane markings. Additionally, we introduce `Vehicle Reflection`,

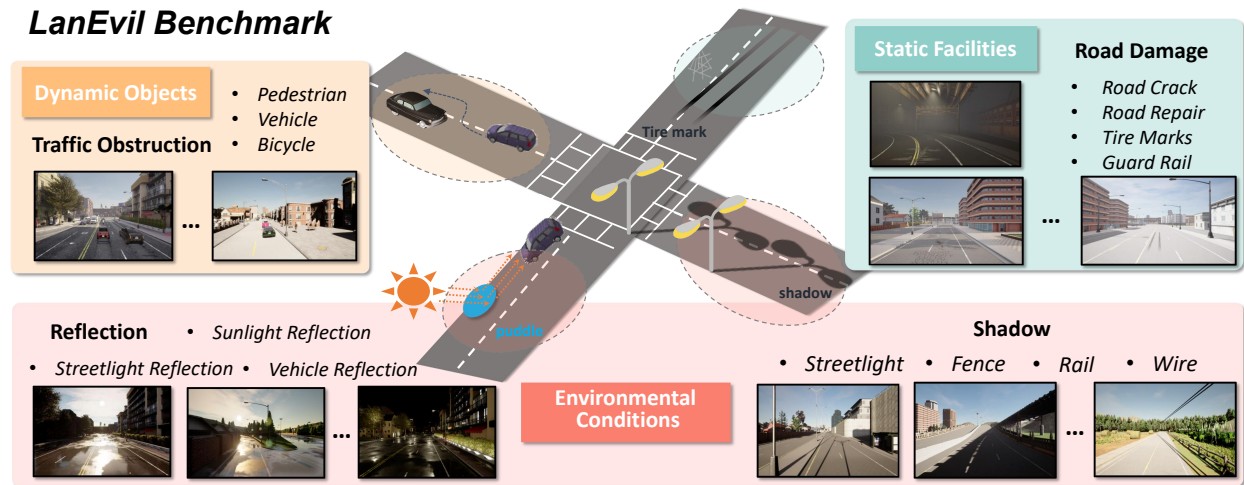

Figure 2: The framework of our *LanEvil* benchmark, which contains 14 specially-designed environmental illusion types from 4 categories including road damage, traffic obstruction, reflection, and shadow.

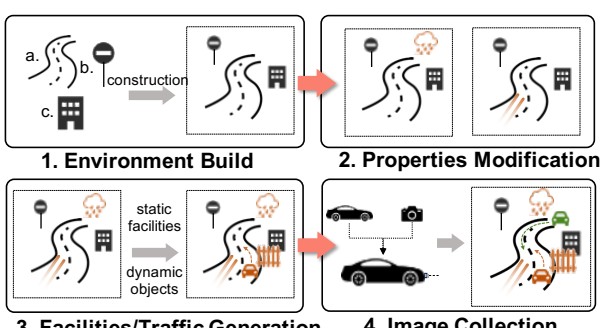

Figure 3: Illustration of our data collection pipeline.

which denotes the reflection of the surface of some vehicles, affecting the visibility of the front vehicles.

## 3.3 Construction Details

**Data collection.** Though the proposed environmental illusions naturally exist, it is difficult to directly collect large-scale real-world images containing diverse types of illusions. Therefore, we use CARLA [13] simulator to generate high-quality perturbed cases and then sample the images with high visual fidelity. The data collection pipeline is as follows: ❶ according to real-world scenarios, we customize the 3D environment and design road types (map segmentations) that are commonly witnessed for traffic, such as T-junction; ❷ we perturb the properties of specific objects in the simulation environment based on the illusion generation methods in Section 3.2; ❸ we place objects in different positions and generate traffic flow; ❹ we run our vehicle agent under the given routing path, and then save the case and capture the first-view images. The sampling RGB camera is positioned in front of the vehicle agent with 1280 × 720 resolution and 90.0° field of view. To make it more practical, we follow [5, 14] and replicate the most basic and common traffic cases for autonomous driving such as following other vehicles and executing sharp turns. The pipeline and collected images are shown in Figure 3 and 5, respectively.

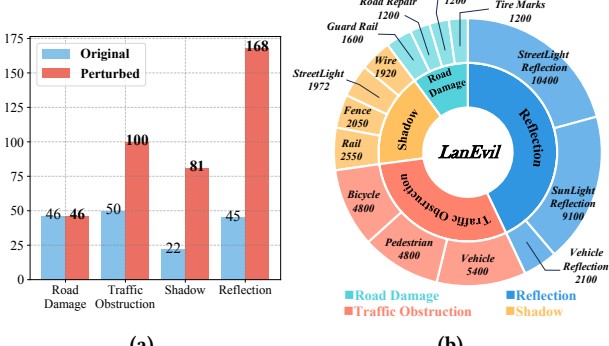

**(a)**                    **(b)**

Figure 4: The statistics of *LanEvil* dataset. (a) The number of original and perturbed cases under four categories. (b) The case distribution of four illusion categories.

**Quality control.** We follow a similar annotation quality control procedure to classical datasets [31, 55]. Here, all of our annotators followed the same annotation guidelines, including what to annotate and how to annotate lanes. Moreover, to ensure the accuracy of annotation, we divide the annotators into 3 groups. Each image was assigned to 2 groups for annotation. Then the average results were reviewed by the third group.

## 3.4 Data Properties

**Subset division.** Our *LanEvil* contains two subsets, *i.e.*, a training set with normal images and a test set with environmental illusions. We first generate the training set with 40,000 randomly sampled clean images without designed environmental illusions. The test set consists of 50,292 images. For each basic environmental illusion, we provide an original case without any illusion and 2 - 10 perturbed cases, each consisting of 50 to 300 consecutively captured driving images. The statistics of original and perturbed cases are shown in Figure 4a and Table 1b. *More dataset details including license can be found in Supplementary Material.*

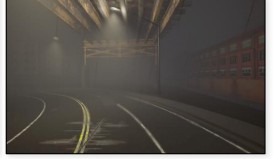 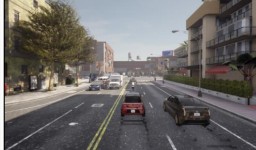 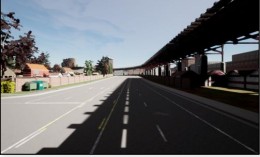 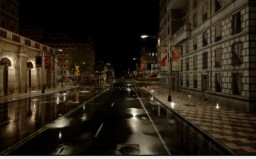

(a) Origin    (b) Road Damage    (c) Traffic Obstruction    (d) Shadow    (e) Reflection

**Figure 5: Visualization of images from our *LanEvil* dataset under different environmental illusions.**

**Table 1: Detailed data properties of *LanEvil*.**

**(a) Scenario diversity**

| Type | Number | Typical examples |
|------|--------|------------------|
| Scene | 5 | Urban, Highway |
| Lane line | 9 | White single solid, Yellow double solid |
| Weather | 12 | SoftRainNoon, ClearNoon |
| Road type | 9 | T-junction, Uphill |

**(b) Quality distribution**

| Image Type | Road Damage | Traffic Obstruction | Shadow | Reflection |
|------------|-------------|---------------------|--------|------------|
| Original | 2,600 | 5,000 | 2,051 | 4,600 |
| Perturbed | 2,600 | 10,000 | 6,441 | 17,000 |
| Total | 5,200 | 15,000 | 8,492 | 21,600 |

**Category distribution.** Our *LanEvil* test set contains 50,292 images comprising a total of 94 basic cases (*e.g.*, straight ahead, turn) with editable 3D environments. The quantity of cases and images for each type of environmental illusion is illustrated in Figure 4b. In addition, The *LanEvil* dataset encompasses 9 line types in different shapes and colors. It also covers multiple driving scenes and multiple road types, as shown in Table 1a.

**Other application possibilities.** Besides image collection, our dataset also involves the meticulous construction of large-scale corrupted 3D simulation scenarios. These scenarios are saved in editable formats which could support further development; in addition, these dynamic scenarios could be used as input to other software for evaluation, since CARLA has been successfully connected to many other systems.

## 4 ATTENTION AREA MIXING (AAM)

To address environmental illusions, we introduce the Attention Area Mixing here. The framework is shown in Figure 6.

Recent studies, such as Geirhos et al. [17], highlight a significant texture bias in DNNs. Following this, Liu et al. [35] proposed utilizing category-specific features to improve training. Drawing on prior research and [62], we introduce the AAM approach, specifically designed to meet the unique requirements of LD. We create a repository of high-attention (HAA) areas from hard examples, which are then blended with dataset images for augmentation. This method aims to enhance LD model performance significantly.

### 4.1 Attention Area Generation

The attention graph is instrumental in visualizing the regions that the model prioritizes during the prediction phase. By incorporating visual attention mechanisms such as CAM [64], Grad-CAM [50], and Grad-CAM++ [7], we significantly bolster the interpretability

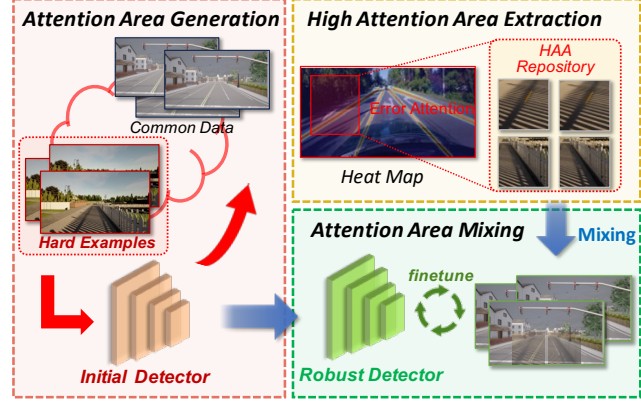

**Figure 6: The Attention Area Mixing (AAM) Framework**

and insight into deep learning models. Our approach meticulously evaluates the precision of the model's focus on relevant sectors by examining its attention map. More precisely, for a given image $I$, we compute its attention map $M$ employing an attention module $\mathcal{A}$, specifically designed for LD:

$$M = \mathcal{A}(I). \tag{4}$$

More precisely, the attention module $\mathcal{A}$ for LD is

$$\mathcal{A}(I) = \text{relu}\left(\sum_k \sum_i \sum_j \alpha_{ij}^k \cdot \text{relu}\left(\frac{\partial \mathcal{L}}{\partial A_{ij}^k}\right) \cdot A_{ij}^k\right), \tag{5}$$

where $\alpha_{ij}^k$ represents the gradient weights for the lane detection task within activation map $k$, $\mathcal{L}(\cdot)$ denotes a loss function optimized for lane detection, $A_{ij}^k$ is the pixel value in position $(i, j)$ of the $k$-th feature map, and $\text{relu}(\cdot)$ denotes the RELU function.

Additionally, our methodology concentrates exclusively on hard examples—images that the model does not readily detect accurately. For any given image, should its Accuracy or F1-score be lower than the dataset's mean, we designate it as a hard example.

### 4.2 High Attention Area Extraction

The process of High Attention Area (HAA) Extraction identifies mismatches between model focus and ground truth. It begins by applying Gaussian blurring to the heatmap $M$ to improve generalization, represented as:

$$M' = G(M, \sigma), \tag{6}$$

with $M'$ as the blurred heatmap, where $\sigma$ is the Gaussian kernel's standard deviation. This step is followed by thresholding $M'$ to produce a binary map $B$:

$$B(x, y) = \begin{cases} 1 & \text{if } M'(x, y) > T, \\ 0 & \text{otherwise,} \end{cases} \tag{7}$$

here, $T$ is the threshold, with $x, y$ as pixel coordinates. We identify connected regions in $B$, remove areas smaller than size threshold $S$, forming interest areas $A$:

$$A = \{r \mid \text{area}(r) > S, r \subset B\}. \tag{8}$$

Discrepancies are pinpointed by comparing each region $r \in A$ with ground truth $G$, defining mismatches as regions with insufficient overlap and extracting the minimum bounding rectangles (MBR) for these regions:

$$\mathbb{R}_{HAA} = \{\text{MBR}(r) \mid r \in A \wedge \text{overlap}(r, G) < \theta\}, \tag{9}$$

where $\mathbb{R}_{HAA}$ represents the set of mismatched regions enclosed by their minimum bounding rectangles, and $\theta$ establishes the threshold for acceptable overlap.

### 4.3 Attention Area Mixing

Following the creation of the $\mathbb{R}_{HAA}$, we perform mixed operations to infuse the dataset with hard examples. Given the dataset $\mathbb{D}$, for each image $I$ within $\mathbb{D}$, we integrate a randomly selected High Attention Area $H$ from $HAA$ Repo into $I$, through the operation:

$$I_{\text{mixed}} = I \oplus \text{Locate}(H, G_I), \tag{10}$$

here, $I_{\text{mixed}}$ signifies the augmented image, incorporating $H$ based on the ground truth $G_I$, with $\oplus$ representing the blending action. This method enhances model resilience to real-world variability by embedding critical High Attention Areas into training images, fostering improved recognition and detection accuracy.

## 5 EXPERIMENTS

### 5.1 Experimental Setup

**Dataset.** In our main experiments, we first train the LD models from scratch using the training set of our *LanEvil*, and then evaluate their robustness on the test set of *LanEvil*. We also evaluate the domain gap between simulated data and real data in Section 5.4.

**Target models.** To provide a comprehensive evaluation, we choose five representative and commonly-used LD models from the five LD categories as introduced in Section 2.1 for experiments: LaneATT [53], UltraFast [42], BezierLaneNet [15], GANet [57], SCNN [41]. For the backbones, we use the ResNet [19] series including ResNet-18, ResNet-34, ResNet-50, and ResNet-101 with weights pre-trained on ImageNet [10]. Note that, the official SCNN model implementation only supports VGG-16 [51] architecture. Therefore, we only use VGG-16 as its backbone.

**Evaluation metrics.** We select the two most widely used metrics in lane detection, *i.e.*, *Accuracy* and *F1-score*, as the main evaluation metrics to calculate the performance of LD methods. For both of these metrics, *higher* values indicate *better* performance/robustness. *Detailed definitions can be found in the Supplementary Materials.*

### 5.2 Main Results

We first evaluate the model robustness of five LD models on *LanEvil*. Due to the space limitations, we report the average performance

of models on each of the four main illusion categories here. *The breakdown results of each illusion type and level can be found in the Supplementary Material.* As shown in Table 2, we can **identify**:

❶ Overall, the proposed environmental illusions have demonstrated certain impacts on the robustness of LD models. In general, these illusions can cause an average absolute **5.37%** Accuracy drop and **10.70%** F1-Score drop.

❷ Different environmental illusions show different threat impacts on LD model robustness. For instance, Shadow demonstrates the most pronounced effect, leading to an average model performance decrease of 7.39%; in contrast, Road Damage shows comparatively weak influence with only 2.49% decreases on average.

❸ Following a comprehensive analysis of the model and backbone, we observe that different models display various degrees of resistance to these types of corruption. In particular, GANet showcases the highest clean performance. However, the Accuracy decreases the most after attacks, amounting to approximately 7.53%. Conversely, SCNN has the lowest Accuracy in clean images but experiences the least Accuracy decrease. Additionally, in terms of model backbones, as the depth of layers increases, the performance and robustness of the model tend to improve.

❹ As the severity level increases, the performance degeneration increases significantly. Specifically, the level-1 images cause an average 2.61% Accuracy drop and 7.06% F1-Score drop; in contrast, the level-5 images can cause an average **19.12%** Accuracy drop and **34.10%** F1-Score drop.

### 5.3 Results on Noise Defense Methods

In this section, we further investigated the effectiveness of our proposed AAM method and existing noise defense methods on our *LanEvil* dataset. Specifically, we choose ResNet-18 as the backbone for the LaneATT and UFLD models and apply PGD adversarial training [38], cutout [11], copy-paste [18], Augment HSV and MixUP [62] for the noise defense methods as they improve the model robustness towards adversarial noises [32–35] or natural corruption.

Figure 7 demonstrates that, across various noise defense methods, our AAM leads with a notable 3.76% average increase in accuracy, outperforming competing methods. Other defense strategies yield only modest gains, typically below 1.5%, with an overall average improvement of +1.24%. Notably, PGD-AT experienced performance declines compared to vanilla models (-1.45%). These outcomes suggest that the environmental illusions introduced by our method diverge from conventional adversarial noise and corruption, highlighting the need for specialized defense studies.

### 5.4 Visual Fidelity Analysis

In this part, we further study the visual fidelity of our generated corrupted images. Specifically, we conduct two experiments: (1) training on either simulated or real-world datasets and then testing on another dataset; (2) conducting human perception studies on the visual quality of our dataset.

**Cross-domain model prediction.** ❶ *LanEvil → Real-world.* For each of the three real-world datasets (TuSimple, CULane, LLAMAS), we separately train a model on the original real-world dataset, a model on the generated *LanEvil*, and a *LanEvil* pre-trained model fine-tuning on 100 images from the corresponding real-world datasets.

**Table 2: Evaluation results of different LD models using ResNet-18 on the *LanEvil* dataset. LD models are trained using the *LanEvil* training set. For each category of illusion, we report the average value over different types and severity levels. The bold values represent the minimum in each column, and "Gap" is computed by "Perturbed" minus "Original". *More results of other backbones and illusions breakdown can be found in Supplementary Material.***

**(a) Results under Accuracy (%)**

| Method | Road Damage | | | Traffic Obstruction | | | Shadow | | | Reflection | | | *Average* | | |
|---|---|---|---|---|---|---|---|---|---|---|---|---|---|---|---|
| | Perturbed | Original | Gap | Perturbed | Original | Gap | Perturbed | Original | Gap | Perturbed | Original | Gap | Perturbed | Original | Gap |
| LaneATT [53] | 76.23 | 78.66 | -2.43 | 73.18 | 75.84 | -2.66 | 79.48 | 86.07 | -6.59 | 71.33 | 78.93 | -7.59 | 73.53 | 78.85 | -5.32 |
| UFLD [42] | 65.90 | 68.47 | -2.57 | 65.64 | 67.58 | -1.94 | 67.73 | 74.01 | -6.27 | 64.45 | 70.88 | -6.42 | 65.45 | 69.90 | -4.44 |
| BezierLaneNet [15] | 73.14 | 75.76 | -2.62 | 71.44 | 74.51 | -3.07 | 68.79 | 78.32 | -9.54 | 69.44 | 75.31 | -5.87 | 70.00 | 75.52 | -5.52 |
| GANet [57] | 85.32 | 89.33 | **-4.01** | 80.53 | 84.75 | **-4.22** | 83.44 | 93.02 | **-9.58** | 79.21 | 89.23 | **-10.02** | 80.55 | 88.08 | **-7.53** |
| SCNN [41] | 71.11 | 71.94 | -0.84 | 67.17 | 69.67 | -2.50 | 65.38 | 70.33 | -4.95 | 63.88 | 68.05 | -4.16 | 65.31 | 69.36 | -4.05 |

**(b) Results under F1-score (%)**

| Method | Road Damage | | | Traffic Obstruction | | | Shadow | | | Reflection | | | *Average* | | |
|---|---|---|---|---|---|---|---|---|---|---|---|---|---|---|---|
| | Perturbed | Original | Gap | Perturbed | Original | Gap | Perturbed | Original | Gap | Perturbed | Original | Gap | Perturbed | Original | Gap |
| LaneATT [53] | 49.20 | 52.63 | -3.43 | 46.03 | 51.87 | -5.84 | 48.23 | 60.35 | -12.12 | 38.81 | 53.23 | -14.41 | 42.95 | 53.80 | -10.85 |
| UFLD [42] | 21.25 | 24.54 | -3.30 | 28.23 | 31.85 | -3.62 | 26.18 | 36.32 | -10.13 | 19.96 | 30.82 | -10.85 | 23.57 | 31.65 | -8.08 |
| BezierLaneNet [15] | 37.84 | 40.68 | -2.85 | 42.53 | 50.03 | -7.49 | 41.41 | 50.93 | -9.52 | 37.06 | 45.86 | -8.81 | 39.49 | 47.95 | -8.46 |
| GANet [57] | 79.01 | 84.16 | **-5.15** | 70.34 | 80.79 | **-10.45** | 73.50 | 89.33 | **-15.83** | 65.25 | 83.03 | **-17.78** | 68.64 | 83.25 | **-14.61** |
| SCNN [41] | 48.69 | 52.88 | -4.19 | 43.63 | 50.47 | -6.85 | 34.92 | 49.42 | -14.50 | 34.70 | 46.94 | -12.24 | 37.71 | 49.19 | -11.49 |

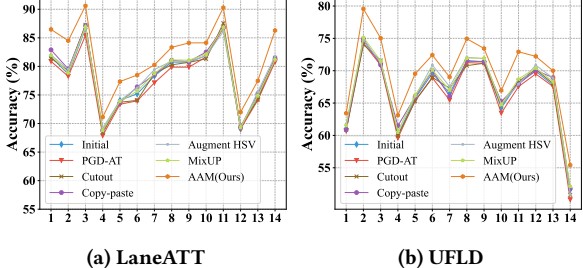

**(a) LaneATT**   **(b) UFLD**

**Figure 7: Evaluation of noise defense methods. The designed illusions are resilient to the data defense methods employed. The x-axis indicates different types of environmental illusions. *More details can be found in Supplementary Material.***

All the models are LaneATT with ResNet-34. Here, we report the F1-score (%) on the testing set of TuSimple (96.77, 91.56, 94.23), CULane (76.68, 69.32, 73.98), and LLAMAS (93.74, 89.15, 92.58). The results indicate acceptable domain gaps despite differences between the real world and our simulated images. Additionally, the gap can be narrowed by incorporating a small number of real images. ❷ *Real-world → LanEvil.* Furthermore, we use LaneATT models with ResNet-34 and train them on real-world datasets (*i.e.*, TuSimple, CULane, and LLAMAS), and then w/ or w/o fine-tuning them on the *LanEvil* . For each model, we test it separately on corresponding real-world datasets and *LanEvil* . Table 3 shows minor performance reductions, indicating a comparatively small domain gap between *LanEvil* and other real-world datasets.

**Human perception study.** Following [29], we conduct human perception studies and ask the participants to evaluate the naturalness of the collected 300 images (150 from *LanEvil* and 150 from real-world scenarios). Specifically, We recruited 100 participants from campus, all with normal (corrected) eyesight. For each image, participants first view it for 3 seconds, and then rate the image by a 5-point Absolute Category Rating (ACR) [22]. All participants

**Table 3: F1-Scores (%) of LaneATT with ResNet-34 w/o and w/ fine-tuning on *LanEvil* training set.**

| Dataset | Real-world test set | | | *LanEvil* test set | | |
|---|---|---|---|---|---|---|
| | w/o | w/ | Gap | w/o | w/ | Gap |
| TuSimple [55] | 96.77 | 95.40 | -1.37 | 48.34 | 62.92 | +14.58 |
| CULane [41] | 76.68 | 76.14 | -0.54 | 55.98 | 68.41 | +12.43 |
| LLAMAS [4] | 93.74 | 93.09 | -0.65 | 58.26 | 66.95 | +8.69 |

are asked to finish the evaluation in 30 minutes. The average ACR results (our images: 3.89 and real-world images: 3.98) suggest that our simulated images are comparatively natural to human vision when compared to real-world images.

## 6 EVALUATION ON COMMERCIAL SYSTEMS

Here, we conduct software-in-the-loop tests on two commercial autonomous driving systems including OpenPilot and Apollo. These systems contain perception and decision modules, which have been applied in real-world auto-driving vehicles (*e.g.*, TOYOTA, Baidu Apollo). To conduct experiments, we directly feed the 3D cases in *LanEvil* as the perception input for the systems and evaluate their final decision performance.

### 6.1 OpenPilot Simulation

We first evaluate *LanEvil* on OpenPilot, an open-sourced commercial driver assistance system that provides a range of functions. We chose eight cases (two types under each category of illusion) in *LanEvil* and configured their starting points and directions to ensure they traverse the road sections we have designed. The pipeline for implementing joint simulation is: ❶ we connect OpenPilot with the source-compiled CARLA 0.9.14, and specify CARLA's initialization parameters (*e.g.*, maps); ❷ for each case pair (clean and perturbed CARLA 3D cases), we select the vehicle model (*i.e.*, Audi)

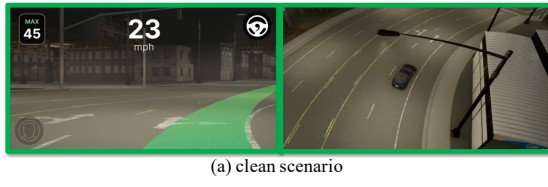

(a) clean scenario

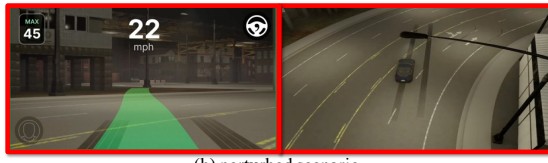

(b) perturbed scenario

**Figure 8: The `Tire Marks` case causes OpenPilot to incorrect decisions leading to collisions on the wall.**

and set the starting point; ❸ we start up the auto-driving mode with a speed limit of 45 mph and observe/evaluate the decision performance. Note that, the starting point is positioned with a 5-10 *meters* distance away from the perturbed location.

We repeat the experiment five times for all eight cases and report the average results. To quantify the results, we follow [48], and use the Attack Success Rate (ASR) as the evaluation metric. Following the US traffic policy [40], the criterion determines an attack or perturbation is successful (*i.e.*, the car is deceived) when it achieves over 0.285m lateral deviations within the required success time (2.5 seconds). As the driving time from the starting point increases, we observe that the ASR for all types of illusions significantly increases. In other words, the auto-driving vehicle turns to make incorrect decisions. In particular, for 92.31% of the frames, the `Road Damage` can make OpenPilot achieve over 0.285m lateral deviations within 2.5 seconds. *More results can be found in Supplementary Material.* We also provide a visualization of the `Tire Marks` cases in Figure 8, where nearly all frames of the OpenPilot system encounter recognition errors, resulting in decision-making mistakes and ultimately leading to car collisions on the wall. The above results indicate that our proposed environmental illusions present certain impacts on the robustness of commercial autonomous driving systems.

## 6.2 Apollo Simulation

We also evaluate *LanEvil* on an auto-driving software platform Apollo developed by Baidu. The evaluated cases are similar to Open-Pilot. However, the evaluation protocol is different since Apollo will stop when encountering the lane lines. Therefore, we measure the percentage of cases where Apollo stops. Also, different from the pipeline in Section 6.1, in step ❸, we simultaneously select the start point and end point of the vehicle for each case. During the experiments, the Apollo vehicle stopped in 6 cases over the 8 tests, demonstrating its comparatively weak robustness towards environmental illusions.

To sum up, the above studies on two systems demonstrate the potential threats of our proposed environmental illusions to commercial autonomous driving systems, which yield strong safety concerns on real-world auto-driving vehicles. *More visualizations and details can be found in Supplementary Material.*

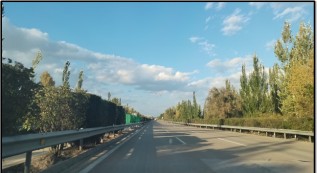
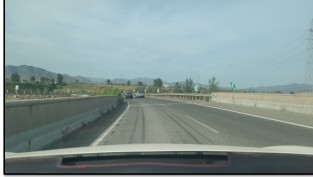

(a) Fence Shadow    (b) Tire Marks

**Figure 9: Real-world environmental illusion images.**

## 7 REAL-WORLD CASE STUDIES

Finally, we extend our experiments from the simulation environment to the real-world images (as shown in Figure 9) to verify the threats of the proposed environmental illusions in real-world scenarios. Specifically, a stationary camera was mounted on a vehicle to acquire video recordings from various highways and urban routes. These routes were repeatedly navigated under different weather and time conditions to capture a wide range of environmental illusions. Subsequently, we meticulously select 1,400 images that cover 14 types of illusions as the perturbed set from all collected frames and select the corresponding normal scene frames as the clean set. All images were subject to manual annotation for precise analysis.

We use TuSimple pre-trained LaneATT with ResNet-34 to report the Accuracy drop (%) on `Road Damage` (-7.48%), `Traffic Obstruction` (-8.61%), `Shadow` (-13.55%), and `Reflection` (-12.84%). The tendency of the impact of illusions on real-world images stays the same with simulator conclusions basically, where `Shadow` demonstrates the most pronounced effect while `Road Damage` shows comparatively weak influence. In particular, the main observations in the simulator hold for real-world images, showcasing an even more pronounced impact, which indicates that environmental illusions also pose a significant threat in the real world.

In addition, we drove a real car and turned on the assistant-driving mode in the real-world shadow illusion scenario, where we identified incorrect decisions with noticeable steering deviation. Since the vehicle has not been released, we are bound by the confidentiality agreement and cannot disclose more details. However, we report a demo video on the website. The above results demonstrate that the proposed environmental illusions also have high risks in the real world, which requires wider attention in the future.

## 8 CONCLUSIONS AND FUTURE WORK

This paper studies the potential threats caused by the environmental illusions to LD models and establishes the first comprehensive benchmark *LanEvil* for evaluation. Large-scale experiments on *LanEvil* demonstrate that those naturally existing environmental illusions significantly reduce the performance of LD models, which necessitates further attention for building robust auto-driving systems. We will release our dataset upon paper publication.

**Limitations.** Despite the promising results, there are several directions/limitations we would like to explore: ❶ evaluation of *LanEvil* on end-to-end foundation models for autonomous driving; ❷ evaluation of *LanEvil* on other common tasks in autonomous driving such as 3D obstacle detection; ❸ extending the size of real-world images with environmental illusions; and ❹ testing more product-level real-world autonomous driving vehicles.

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
