# OpenReview forum: "LanEvil: Benchmarking the Robustness of Lane Detection to Environmental Illusions"
_acmmm.org/ACMMM/2024/Conference — MM2024 Poster_

### Official Review · Reviewer_EWLP · 2024-05-12

**Rating:** 6
**Confidence:** 3

**Summary:**

Amazing! Thanks to the researchers, this study aims to solve a real and difficult bottleneck in lane detection - no obvious visual information.

**Strengths:**

1. This paper proposes a new dataset benchmark to test the robustness of lane detection networks;
2. This paper studied the influence of commonly used data enhancement methods on the classic LaneATT and UFLD models. It proposed a data enhancement method more suitable for lane detection scenarios.

**Limitations:**

1. The experimental methods in this paper need to be updated. The anchor-based method CLRNet has shown strong performance in many articles this year (except ADNet), and the performance of UFLDv2 in row-based is greatly improved compared with UFLD.

**Suitability:**

2

---

### Official Review · Reviewer_nj4u · 2024-05-15

**Rating:** 5
**Confidence:** 2

**Summary:**

The paper presents LanEvil, a new benchmark aimed at evaluating the robustness of lane detection (LD) models against environmental illusions such as shadows and tire marks. These illusions, which are often overlooked in current benchmarks, can significantly impact the performance of LD in real-world scenarios. The authors have designed 14 types of environmental illusions and created a dataset of 90,292 images using the CARLA simulator. Extensive experiments with popular LD methods reveal performance degradation when faced with these illusions, with the Attention Area Mixing (AAM) approach showing promise in improving robustness.

**Strengths:**

1. Clarity and Motivation: The paper is well-written, with a clear presentation of the motivation behind the study and a comprehensive description of the dataset design.
2. Visualization: The figures and visualizations are clear and effectively support the understanding of the benchmark and the results.
3. Validation and Novel Approach: The authors have validated the benchmark with multiple lane detection methods and introduced the AAM method to enhance robustness, demonstrating its effectiveness.

**Limitations:**

1. Academic Contribution: While the work is substantial and contributes to the robustness of lane detection benchmarks, the core dataset generation relying on CARLA may limit the academic novelty. The use of simulation, though practical, might not fully capture the complexity of real-world conditions.
2. Domain Gap Validation: The paper claims acceptable domain gaps despite differences between real-world and simulated images. However, it lacks comparative results from other simulation datasets of similar scale, which could have further validated the generalizability and robustness of the proposed benchmark.

**Suitability:**

2

---

### Official Review · Reviewer_prbt · 2024-05-17

**Rating:** 4
**Confidence:** 4

**Summary:**

This paper is the first to explore the robustness of autonomous driving lane detection (LD) systems against environmental illusions such as shadows and tire marks. It establishes the first comprehensive benchmark, LanEvil, to assess LD's resistance to these natural disturbances. Using the CARLA simulator, 94 realistic cases were created, resulting in a dataset of 90292 images. Experiments show a significant performance drop for popular LD methods when faced with these illusions, while the proposed Attention Area Mixing (AAM) method significantly enhances robustness under lighting effects.

**Strengths:**

Novelty: This paper introduces the LanEvil dataset for assessing the robustness of Lane Detection (LD) against environmental illusions, offering a novel perspective in this field. The dataset includes 14 different types of environmental illusions that cover most factors in the real world that could affect LD tasks, such as Road Damage, Traffic Obstruction, Shadow, and Reflection. By creating these Corner case scenarios, the paper provides a more comprehensive evaluation angle for mainstream LD algorithms. The LanEvil dataset also provides valuable resources for future research in the field of lane detection, especially in Corner Case scenarios.

Technical Correctness: The design of each type of environmental illusion in the paper has been carefully considered to ensure they can realistically simulate complex real-world situations. Particularly, the simulation of shadows and road damage is very interesting and poses a significant challenge to mainstream LD algorithms, which has been fully verified in Table 2.

Adequate Evaluation: The paper evaluates several popular LD methods on the LanEvil dataset, showing the degree of performance degradation caused by different illusions. These experimental results not only prove the effectiveness of the dataset but also reveal the deficiencies in the robustness of existing LD methods.

**Limitations:**

Overall, this paper provides a novel and interesting perspective for evaluating the robustness of lane detection, which is an interesting work. However, I have the following suggestions for further consideration:

Traffic Obstruction: Although the paper has done some work in this area, it does not have a significant advantage over existing real datasets. In addition, most current works using Carla as the experimental environment set traffic obstructions (vehicles or pedestrians) as a default condition to simulate real traffic environments as closely as possible. It is recommended to utilize Carla's Scenario Runner for creating more realistic simulations of autonomous driving environments, focusing on vehicle and pedestrian scenarios.

Reflection: The treatment of reflections in the paper is relatively conventional and has been widely used in related works based on Carla (different weather, lighting conditions). It is recommended that this part be set as a default setting for data augmentation.

Dataset Distribution: Figure 4 indicates that the current data focuses more on easily collected reflections and traffic obstruction illusions. The paper should concentrate more on environmental factors that significantly affect the integrity of the lane lines and cause "false lane lines" illusions, such as shadows and road damage, to balance the proportion of different illusion categories in the LanEvil dataset.

Cross-Domain Model Prediction: In Section 5, the paper proposes that models trained with the LanEvil dataset show a small performance loss in F1-score when directly evaluated on the TuSimple, CuLane, and LLAMAS datasets. For Sim2Real, models trained in simulation environments usually find it difficult to replicate their performance in real-world data without domain adaptation or similar methods. For example, the paper "Domain Adaptation In Reinforcement Learning Via Latent Unified State Representation" discusses the significant performance degradation when models trained from the source domain of Carla are directly transferred to the target domain.
In addition, it is recommended to include test results of real data-trained models on the "illusion-free" LaneEvil dataset, thereby ablate the performance degradation caused by "illusions" as presented in Table 3.

**Suitability:**

2

---

### Official Review · Reviewer_EXR9 · 2024-05-24

**Rating:** 2
**Confidence:** 4

**Summary:**

This paper addresses the robustness of lane detection (LD) models in autonomous driving systems against environmental illusions such as shadows and tire marks, which are common in real-world traffic situations. Existing benchmarks often neglect these challenges, posing significant safety risks. The paper introduces LanEvil, the first comprehensive benchmark for evaluating LD robustness against these natural corruptions. The authors designed 14 types of environmental illusions and created a dataset with 94 customizable 3D cases and 90,292 images using the CARLA simulator.

Extensive experiments revealed significant performance degradation in popular LD methods when exposed to these illusions, with shadows having the most considerable impact. The paper also assesses commercial systems like OpenPilot and Apollo, showing that environmental illusions can lead to incorrect decisions and potential accidents. To counter these issues, the authors propose the Attention Area Mixing (AAM) approach, which improves robustness by 3.76%.

Key contributions include the introduction of the LanEvil benchmark, the proposal of the AAM approach, and extensive experiments demonstrating the real-world threats posed by environmental illusions.

**Strengths:**

1. The paper uses numerous figures to illustrate its design concepts, making it quite intuitive.
2. The paper provides extensive information in the appendix.

**Limitations:**

1. This paper only utilizes video as the input modality, which does not align with the multimodal theme expected by MM24. It is recommended to consider submitting to a conference focused solely on computer vision.
2. The datasets and methods mentioned in the paper are not provided in supplementary materials or an online GitHub repository, making it difficult for reviewers to verify the reported results. Additionally, there is no mention of any plans for open-sourcing the work.
3. The images in the dataset appear artificial and differ significantly from real-world datasets, raising concerns about the practical significance of the tests conducted on them.
4. The figures and tables in the paper are poorly formatted. For instance, some text in Figure 2 is excessively large, while in Figure 4, it is too small. Figures 2 and 3 are crowded together, as are Figures 5, 6, and Table 1. Similarly, Figure 7 is too close to the text below it. In contrast, there is excessive spacing around Figures 8 and 9 and the accompanying text. This inconsistency in formatting suggests a lack of thorough review by the authors after completing the manuscript.

**Suitability:**

1

---

### Meta-Review · Area_Chair_Lj1r · 2024-06-30

**Recommendation:** Accept (Poster)
**Confidence:** 5

**Metareview:**

This paper proposes a new benchmark aimed at evaluating the robustness of lane detection (LD) models against environmental illusions such as shadows and tire marks, and a new method to mitigate the performance drop under illumination effects. The benchmark is collected in CARLA simulator with extensive design in environmental illusions and scenarios. The proposed benchmark exposes the robustness problems in current LD methods with adequate experiments.

The preliminary ratings are (wr, ba, wa, a). Key concerns are:
1. Logistics issues. Misalignment to the MM theme. Not releasing datasets and methods during submission.
2. Sim2real gap caused by CARLA simulator.
3. Writing and format issues.
4. Experiment concerns. Issues in dataset quality and distribution. Lack of comparison to more advanced methods.
5. Lack of technical contributions.

After author rebuttal, the final ratings are (wr, ba, na, wa), na for not-available. Reviewer nj4u does not provide their final rating, and reviewer EWLP lower their rating from a to wa. AC stepped in, read the paper, review comments, the rebuttal and the reviewer's final justification. In the view of AC, reviewers are focusing on concerns 2&4, while author gives more experiment addressing the sim2real test gap with some analysis. Concern 1 is not standing firmly in the view of AC. And concern 5 is not holding up as this paper does provide valid information to the community via its substantial work.

Based on the comments above, AC decides to accept the paper since the paper shows significant improvement thanks to the addressed concerns. The authors should improve the manuscripts according to the comments above.